# Conservation prioritization can resolve the flagship species conundrum

Jennifer McGowan [1,2,3✉], Linda J. Beaumont [1], Robert J. Smith [4], Alienor L.M. Chauvenet [2,5], Robert Harcourt [1], Scott C. Atkinson[2,6], John C. Mittermeier[7], Manuel Esperon-Rodriguez [1,8], John B. Baumgartner [1,9], Andrew Beattie[1], Rachael Y. Dudaniec [1], Richard Grenyer[7], David A. Nipperess [1], Adam Stow [1] & Hugh P. Possingham[2,3]

Conservation strategies based on charismatic flagship species, such as tigers, lions, and elephants, successfully attract funding from individuals and corporate donors. However, critics of this species-focused approach argue it wastes resources and often does not benefit broader biodiversity. If true, then the best way of raising conservation funds excludes the best way of spending it. Here we show that this conundrum can be resolved, and that the flagship species approach does not impede cost-effective conservation. Through a tailored prioritization approach, we identify places containing flagship species while also maximizing global biodiversity representation (based on 19,616 terrestrial and freshwater species). We then compare these results to scenarios that only maximized biodiversity representation, and demonstrate that our flagship-based approach achieves 79−89% of our objective. This provides strong evidence that prudently selected flagships can both raise funds for conservation and help target where these resources are best spent to conserve biodiversity.

[1] Department of Biological Sciences, Macquarie University, Macquarie Park, NSW 2109, Australia. [2] Centre for Biodiversity and Conservation Science, University of Queensland, St Lucia, QLD 4072, Australia. [3] The Nature Conservancy, Arlington, VA, USA. [4] Durrell Institute of Conservation and Ecology, School of Anthropology and Conservation, University of Kent, Canterbury, Kent CT2 7NR, UK. [5] Environmental Futures Research Institute & School of Environment and Science, Griffith University, Southport, QLD 4222, Australia. [6] United Nations Development Programme (UNDP), New York, New York, USA. [7] School of Geography and Environment, Oxford University, South Parks Road, Oxford OX1 3QY, UK. [8] Hawkesbury Institute for the Environment, Western Sydney University, Sydney, NSW 2753, Australia. [9] Centre of Excellence for Biosecurity Risk Analysis (CEBRA), School of BioSciences, University of Melbourne, Parkville, VIC, Australia. ✉email: jennifer.mcgowan@tnc.org

Few marketing tools can rally people to support conservation as effectively as those based on flagship species[1], where each species serves as the focus of conservation marketing campaigns based on its possession of traits that appeal to a target audience[2]. The intention of these campaigns is to capitalize on the audience's affinity for wildlife in order to increase the flow of capital for conservation organizations and raise awareness for their projects and programs. More specifically, they are generally used to support two project types: those that target and benefit the species directly, and those that focus on broader issues, such as protecting the land and seascapes where the flagship is found[2]. However, the flagship approach has been widely criticized because organizations frequently choose species based solely on their appearance, favoring colorful and/or large animals with forward-facing eyes because these characteristics appeal most to the broadest audience[3,4]. This often limits the selection of flagships to species with perceived charisma.

This narrow focus has obvious limits when fundraising for projects to directly benefit a species, as it means these charismatic taxa receive the lion's share of funding[5]. Conservation practitioners and researchers have responded by selecting flagship species for their campaigns based on a wider range of physical, ecological, cultural and threat attributes[2,6–8]. Recent research supports this approach, finding that while appealing species are still the most effective at fundraising, even the least charismatic species can be used successfully to raise funds given targeted marketing effort[1].

There has been much less study on the effectiveness of using flagships when fundraising for projects to conserve places. Past research looked at whether selecting places based only on the presence of charismatic species could also represent broader biodiversity. This approach was found to be ineffective, further emphasizing the need for using less well-known flagships in species-focused campaigns[9–11]. However, the bigger question for conservation is whether it is possible to choose places that are important for broader biodiversity that also contain flagship species, given that fundraisers cannot mislead their target audience about how their money will be spent[12]. Here we ask to what degree does requiring the presence of a flagship species influence our ability to achieve place-based conservation objectives at a global scale?

When it comes to identifying priority areas for global conservation, much research has focused on mapping the distribution of biodiversity assets, such as centers of endemism, uniqueness[13–15], biodiversity hotspots[16–18], taxon-specific diversity[19–21], ecosystem services[22] and the last remaining tracts of wilderness[23]. Organizations often adopt these places that best align with their core mission, and use them to promote high-level strategies and coordinate funding from international or multilateral institutions[24,25].

The underlying rationale for place-based approaches is that protecting these locations from threats[26–28] will best deliver benefits to biodiversity[24,29]. However, identifying a place solely on its physical or ecological attributes does not mean it is a conservation priority[29]. This is because the selection is not bound to a well-defined, objective-driven problem, e.g. maximizing benefits or minimizing threats[30,31]. Clearly defined objectives are vital in conservation decision making, as they provide a transparent approach for guiding action at any scale—from organization's choosing new geographies in which to work through to choosing between actions to conserve species at specific sites. Just as importantly, it allows measurement of the trade-offs involved in accounting for additional constraints, and this is particularly relevant here because it is the only way to measure the impact of targeting flagship species when choosing globally important places for conservation investment.

Here we present a global conservation prioritization approach that integrates the potential for fundraising opportunities. We do so by selecting places that maximize a biodiversity objective while simultaneously ensuring that at least one charismatic flagship species is also found in the selected places. For this analysis, we define the objective based on maximizing the number of non-flagship species, hereafter called background species, that could benefit from conservation should an organization choose to invest in a place. Our approach is flexible in that it can accommodate different objectives according to organizational values[32], but is grounded in traditional planning principles that underpin current global conservation policies for representation and complementarity (Supplementary Fig. 1). This guarantees that all measured biodiversity[33], not just biodiversity that co-occurs with desirable species or places, is safeguarded in the prioritization approach[30,34].

To evaluate our integrated approach, we divide the terrestrial realm into an initial set of 10,200 places (100 km × 100 km). We develop eight global planning scenarios based on different combinations of attributes for both candidate flagship species based on threat status[35], and places, based on their presence in a globally unique ecoregion[36], protected area coverage[37] and degree of human impact[27]. We then compare our results to (i) a place-only approach where the sole purpose is to identify places that maximize the number of background species conserved irrespective of flagship fundraising potential, and (ii) a null test based on the random selection of places which was run 100 times for each of the eight scenarios and does not consider flagship species or representation constraints.

Our integrated approach retains 79−89% of the number of background species available for protection across the range of prioritization scenarios. For some scenarios, this value improves up to 97% assuming organizations might choose to act in the top ten locations delivering the highest potential return on investment. These findings provide strong evidence that flagship species do not need to compromise place-based conservation priorities, as long as they are integrated into a prioritization approach based on clear, quantifiable objectives. This allows organizations and private ventures, whose role in conservation continues to grow, to maximize public awareness and attract funding while accommodating important attributes of both species- and place-based conservation that are relevant to their conservation goals.

## Results

**Flagship species and place-based constraints**. We identified 534 species of mammals, birds and reptiles as candidate flagships (Supplementary Data Table 1), of which 338 were Near-Threatened or higher according to the International Union for the Conservation of Nature (IUCN) Red List (Supplementary Table 1). There were 10,200 places that fell within globally unique ecoregions, 3097 of which overlapped with protected areas, 3961 overlapped with regions of low human impact and 1068 overlapped with both protected areas and low human impact. The number of flagship species used in our eight scenarios varied from 207 to 534 species, depending on their presence in the different subsets of places (Supplementary Fig. 2).

**Evaluating flagship impact on meeting objectives**. To measure efficiency in achieving our objective, we compared the number of background species captured from the places prioritized in our integrated approach with that of the equivalent number of places identified from the place-only and random approaches. For example, when we considered threatened flagships and all place-based attributes as constraints on the prioritization (Fig. 1, Scenario h), the integrated approach delivered 6849 background species in 47 places (Table 1). In comparison, the place-only approach delivered 7702 species for the same number of places,

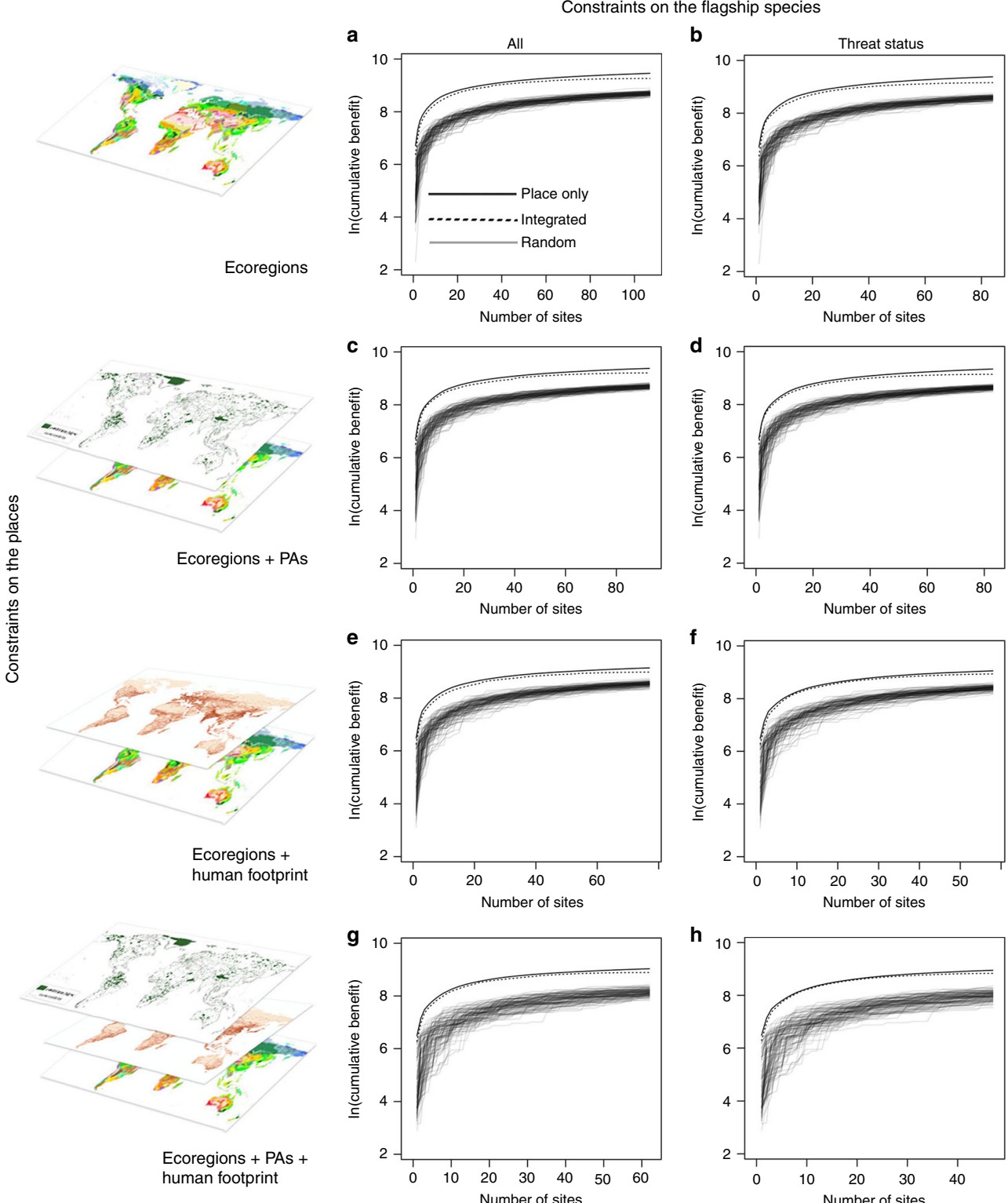

**Fig. 1 Performance comparisons across scenarios.** The scenario performance of the place-only, integrated, and random approaches (**a**–**h**) in achieving biodiversity representation (defined as the number of background species protected). Random selections were performed 100 times for each scenario. Threat status refers to the candidate flagship group subset by IUCN classification of Near-Threatened and higher. Source data are provided in the Source Data file.

**Table 1 Results from the place-only, integrated, and random approaches.**

| Scenarios | Number of background species available | Number of places in place-only solution | Number of places in integrated solution | Number of species in place-only solution | Number of species in integrated solution | Efficiency retained in integrated solution (%) | Efficiency retained in top ten places (%) | Mean efficiency retained in null solutions (%) |
|---|---|---|---|---|---|---|---|---|
| a | 19,616 | 1473 | 107 | 12,878 | 10,545 | 82 | 87 | 47 |
| b | 19,616 | 1473 | 84 | 11,961 | 9487 | 79 | 90 | 45 |
| c | 16,542 | 855 | 93 | 11,835 | 9965 | 84 | 92 | 50 |
| d | 16,542 | 855 | 83 | 11,443 | 9387 | 82 | 92 | 50 |
| e | 12,053 | 554 | 77 | 9362 | 7972 | 85 | 89 | 55 |
| f | 12,053 | 554 | 58 | 8557 | 7621 | 89 | 96 | 52 |
| g | 9833 | 287 | 62 | 8363 | 7269 | 87 | 93 | 41 |
| h | 9833 | 287 | 47 | 7702 | 6849 | 89 | 97 | 39 |

Values describe the maximum number of background species available for each scenario (Scenarios a–h) as described in Table 2 and Fig. 1; and the proportion of background species retained in the integrated approach compared to the place-only and null models. For additional results, see Supplementary Methods.

although the total number of places identified without a flagship constraint was 287 (Table 1). Therefore, we consider the efficiency retained with our integrated approach to be 89% for this scenario. We found that our integrated approach retained 79−89% of the background species delivered by the place-only approach across all scenarios (Table 1; Fig. 1). Since resources are often limited, we might expect organizations to prioritize the subset of these places that contribute most to their conservation objectives; thus, the potential realized efficiency is likely to be higher. For example, we saw efficiencies rise to capture 87−97% of background species available when only the ten most beneficial places within a scenario were considered (Table 1). In all cases, the integrated approach outperformed the random selections, which retained on average 39−55% of background species compared to the place-only approach (Table 1; Supplementary Fig. 3).

The most efficient scenario at representing background species was Scenario h, described above (Fig. 1) and so we shall discuss these results in more detail to outline the specifics of the approach. This scenario delivered 47 ecoregionally unique places that collectively contained 176 candidate flagships: 111 mammal, 53 bird, and 12 reptile species (Supplementary Data Table 2, Supplementary Table 3). The number of flagships emerging in a single place from this solution varied from 1 to 20 with the highest found in the Naga-Manupuri-Chin hills moist forests ecoregion of India, Bangladesh and Myanmar (Supplementary Data Table 2, Site 6). To use an example from our results in China, we can look to the place identified within the Hengduan Shan Conifer Forests (Fig. 2, Place I) that is flanked by the Tibetan Plateau Steppe and Southwest Temperate Forests ecoregions. This place is home to the iconic Giant Panda (*Ailuropoda melanoleuca*) but, in addition to this prominent flagship, other potential flagship species emerged from our analysis including Takin (*Budorcas taxicolor*), Golden snub-nosed monkey (*Rhinopithecus roxellana*) (both pictured in Fig. 2), Snow Leopard (*Panthera uncia*), and the Chinese softshell turtle (*Pelodiscus sinensis*) (Supplementary Fig. 5). Multiple flagships found in a single place provide flexible options for organizations to select species that best reflect their conservation strategies, donor preferences and local conservation interventions.

## Discussion

Our analysis departs from previous research focused on the ability of flagships to be surrogates for broader biodiversity[10,11] by looking at an entirely different research question, namely, to what degree does requiring the presence of a flagship species influence our ability to maximize a conservation objective at a global scale? We have demonstrated that important places for biodiversity can be prioritized while retaining the fundraising

advantages offered by flagship species[38]. This is key because fundraising campaigns use flagships to illustrate the importance of the biodiversity found within priority places[12]; so, in contrast to fundraising campaigns that promote flagships as umbrella species[6], they do not need to constrain investments to places where the flagship species overlaps in space with other important biodiversity. Thus, for example, funds raised from an ecoregion campaign featuring terrestrial flagship species could be spent on conserving freshwater habitats, overcoming a major objection to flagships that emerged from previous research on their surrogacy value[9–11].

Despite our analysis answering a question of real-world relevance for conservation organizations looking to use flagships to ensure continued fundraising opportunities and branding, our intention is not to advocate a suite of flagship species or places for global conservation. Rather, we illustrate how the selection of flagships can be systematic and objective-based, given a set of conservation goals, target audience and marketing strategy, rather than ad hoc or driven solely by perceived species charisma[39]. Organizations can use our approach and then choose the emerging flagships that best align with the local ecological, conservation and social context[25,40], but should remain transparent about what their investments will deliver for local biodiversity. In addition, we expand the number of flagship species beyond those used in previous studies, which often focused on the most famous charismatic megafauna[1]. All of the 534 species we used as candidate flagships are either already popular with the target audience of potential donors in higher income countries, or are similar in size and appearance to these species[3] (Supplementary Data Table 1).

Organizations will invariably have different perspectives on how to define conservation objectives and constraints relative to their institutional values. Alternative objectives, such as maximizing ecosystem services or minimizing species extinction risk, can easily be incorporated into our approach. There are a number of spatially mapped biodiversity assets that could also serve to inform the selection of candidate places. For example, further constraining places to remaining wilderness areas, hotspots of species richness, or climate refugia[24] or places rich in Alliance for Zero Extinction sites and Key Biodiversity Areas[41]. Our approach allows organizations to tailor the problem definition to their values and objectives, moving beyond static asset maps and towards identifying priority places for conservation action by considering them within a properly constructed problem[30]. Importantly, our integrated approach provides a flexible but rigorous mechanism to guide future conservation investments.

Investments will always be tied to actions (e.g. securing specific parcels of land, restoring degraded habitat, tackling invasive

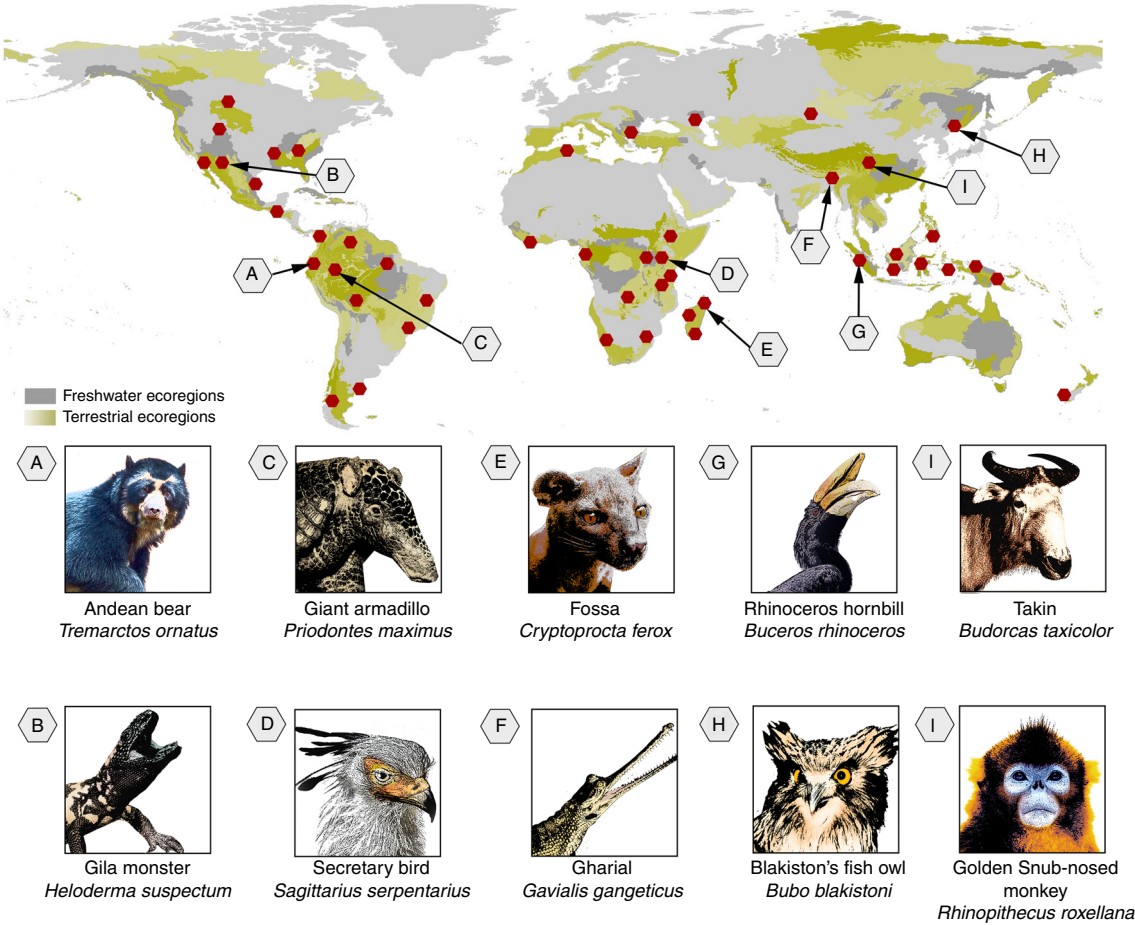

**Fig. 2 Visualization of prioritized places and candidate flagship species.** The map shows the 47 places and a sample of the candidate flagship species (panels A–I) delivered from the integrated approach for Scenario h (Fig. 1h, Table 2). See Supplementary Data Table 2 for full list of species. See Supplementary Table 3 for associated places and ecoregions.

species, establishing an organization in a new landscape, lobbying a government, reintroducing species, etc.). Intended actions should be identified at the beginning of the prioritization, as their associated costs, benefits and feasibility can influence which places emerge as priorities[30] and which species may be best suited to act as flagships. We suggest our approach be considered complementary to more specific systematic conservation planning activities and decision theory approaches, which can further identify the most appropriate placement and timing of management actions at finer scales for practitioners working on multi-species landscape-scale conservation challenges.

Recent estimates suggest the annual budget for realizing global biodiversity goals can reach up to $100 billion per year[42–44]. Given the major and growing role of nongovernment organizations and private ventures in conservation[44,45], we anticipate our approach to be highly relevant to these efforts. Finally, although we examine the role of fundraising potential through flagship species, we recognize that unique places may also resonate with different target audiences and provide a unique marketing platform through which to leverage funds[46]. We conclude that the key to selecting flagship species for fundraising is to move beyond arbitrary selection and apply strategies that confer clear, measurable conservation objectives.

## Methods

**Selecting candidate flagship species.** We used two approaches to identify plausibly charismatic candidate species. For mammals, we used existing

conservation flagships ($N = 80$) and the Cinderella species ($N = 183$) identified by Smith et al.[3]. Cinderella species have similar physical characteristics to flagships, namely large body size and forward-facing eyes, but are not known to be conservation flagships[3]. For reptiles and birds, we identified candidate species using an approach developed by Roll et al.[47] that quantifies interest in species based on their online popularity, measured via the number of Wikipedia page views for a given year. Popular reptile species were taken from the previous work of Roll et al.[47]. Bird species were similarly identified matching the global species taxonomy of the International Ornithological Committee (IOC World Bird List version 7.1) to English language Wikipedia pages and extracting views to each species page for the period between 1 January 2016−1 February 2017 (Mittermeier et al., unpublished). The top 100 reptile and 500 bird species, measured by total page views, were identified as the potential flagship representatives of these groups.

These candidates were further refined to include only species whose ranges have been mapped and made publicly available by the International Union for the Conservation of Nature (IUCN) for mammals and reptiles[35], and BirdLife International and the Handbook of the Birds of the World for candidate bird species[48]. This resulted in a list of 534 species of mammals, birds, and reptiles that we used as the first species attribute class (Supplementary Table 1). These species were then classified according to their IUCN Red-List status (www.iucnredlist.org). We considered taxa classified as Near-Threatened or higher to be in need of conservation action and treated this as a second species attribute class, which reduced the list of candidate flagships from 534 to 338 species globally. We assumed that all species in the list of candidates have equal capacity to serve as a conservation flagship given dedicated marketing efforts[1].

**Background species.** In addition to mammals, birds and reptiles, background species comprised all freshwater crustaceans, carnivorous insects, and amphibians for which IUCN distribution polygons exist in our scenarios (i.e. $N = 19,616$). For all species ranges, we followed an existing approach of Butchart et al.[49] and retained those parts of their distributions marked as either native or re-introduced, and with presence coded as extant, possibly extant, or possibly extinct. We created a presence−absence matrix for both the candidate and background species based

**Table 2 Scenario constructions with species and place-based constraints.**

| Scenarios | Candidate place constraints | Number of places | Candidate flagship constraints | Number of candidate flagships |
|---|---|---|---|---|
| a | G200 Ecoregions | 10,200 | All | 534 |
| b | G200 Ecoregions | 10,200 | IUCN threat status | 338 |
| c | G200 Ecoregions + Protected Areas | 3097 | All | 494 |
| d | G200 Ecoregions + Protected Areas | 3097 | IUCN threat status | 295 |
| e | G200 Ecoregions + Human Footprint | 3961 | All | 447 |
| f | G200 Ecoregions + Human Footprint | 3961 | IUCN threat status | 247 |
| g | G200 Ecoregions + Protected Areas + Human Footprint | 1068 | All | 402 |
| h | G200 Ecoregions + Protected Areas + Human Footprint | 1068 | IUCN threat status | 207 |

The different combinations of place and species attributes used to create the eight scenarios (a−h). Details include the number of sites and candidate flagships available at the beginning of each prioritization. See Supplementary Fig. 2 for distributions.

on the intersections of their ranges with the planning areas used in the different scenarios. Given that many species occupy ranges much smaller than our 100 km × 100 km planning unit size, we erred on the side of caution and did not assign a minimum size threshold to reflect species presence. The coarse resolution of global species range maps means our analysis is subject to errors of omission and commission[50]. However, previous research shows that range maps provide good estimates to inform biodiversity priorities at global scales, but should be combined, when possible, with local data before finer-scale conservation decisions are made[51]. All geoprocessing of spatial data was completed using PostGIS2.3 and ArcGIS v10.3 (ESRI, Redlands).

**Place-based attributes**. We created a global terrestrial equal area planning grid at a spatial resolution of 100 km × 100 km, covering the WWF Global 200[36], a set of terrestrial and freshwater ecoregions identified for their exceptional biodiversity. Each place was then assigned to a unique ecoregion. In instances where more than one ecoregion fell within a place, assignment went to the ecoregion with the largest proportional coverage. In places where terrestrial ecoregions overlapped with larger freshwater ecoregions, we preferentially assigned to the finer resolution terrestrial ecoregions.

For each place, we assigned additional attributes related to protected areas (PA) and the Human Footprint Index (Fig. 1). For PAs, we identified the proportion of each cell allocated to any category of IUCN Protected Areas based on the World Database of Protected Areas[37]. We removed PAs whose status was identified as Proposed, but retained those listed as Not Reported. We used a threshold of ≥10% and ≤90%, as cells with few areas protected (≤10%) might be difficult to establish protection for due to high transaction costs, while cells with considerable coverage (≥90%) might be too spatially constrained to feasibly add more protection. We recognize that these upper and lower bounds are arbitrary, reflecting perceived feasibility; thus, we included additional scenarios where this criterion does not influence the prioritization.

For the Human Footprint Index, we used the mean value calculated for each place[27] and included those places meeting a threshold of <4, as this value reflects landscapes that have not transitioned to be human dominated[52] (Table 2).

**Scenarios**. Based on different combinations of the species- and place-based attributes, we developed eight integrated global planning scenarios (referred to as a −h (Table 1; Fig. 1)). Each scenario was subjected to the customized integrated approach and the place-only approach (Supplementary Fig. 1, Source Code file). For the random tests, we examined how many background species were retained for the same number of sites identified in the integrated approach. No ecoregional representation constraint was considered in the algorithmic decision tree for the random tests. A second species-based randomization test can also be found in the Supplementary Methods. We ran 100 randomizations for each scenario (Fig. 1, Supplementary Figs. 3, 4 and 6, Supplementary Table 2). Results for all approaches and scenarios can be found in the Source Data files. All programming was developed in the R programming language[53].

## Data availability

The spatial datasets used to support the findings of this study are available for download by request from their respective providers. Species range maps can be requested from the IUCN Red List of Threatened Species database found at https://www.iucnredlist.org/resources/spatial-data-download. Protected areas can be requested from the World Database on Protected Areas found at https://www.unep-wcmc.org/resources-and-data/wdpa. WWF Global 200 Ecoregions can be downloaded from Data Basin: https://databasin.org/maps/7c6012cd4026493585483db7b56ff59c. The Human Footprint can be downloaded from NASA's Socioeconomic Data and Applications Center (SEDAC): http://sedac.ciesin.columbia.edu/data/set/wildareas-v3-2009-human-footprint/data-

download. The source data underlying Fig. 1 and Supplementary Fig. 6 are provided as a Source Data file.

## Code availability

The code developed for the integrated prioritization approach can be accessed in the Supplementary files. Code was written by A. Chauvenet and J. McGowan in the R programming language.

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

## Acknowledgements

The authors would like to thank Uri Roll and Shai Meiri for providing the species list for candidate reptiles and John Armanini for rendering the species profiles in Fig. 2. Funding for this research was provided to the Department of Biology at Macquarie University by WildArk.org.

## Author contributions

This project was conceptualized by J.M., L.J.B., R.H., M.E.-R., A.B., R.Y.D., D.A.N., A.S., R.J.S. and H.P.P. Data were sourced by J.M., S.A., R.J.S., R.G. and J.C.M. Data were processed and analyzed by J.M., A.L.M.C., S.A., and J.B.B. under the direction of H.P.P. All authors contributed to the writing and editing of the manuscript.

## Competing interests

The authors declare no competing interests.
