## [Peer Review File · Nature Communications]

Reviewers' comments:

Reviewer #1 (Remarks to the Author):

This ms aims to assess whether flagship species, which are useful for attracting conservation funds, are also good surrogates for other species when identifying priority areas for conservation in a conservation planning framework. The question is important, because if flagships are also good surrogates, then the money spent for their conservation also benefits other species threatened with extinction. Although several questions around flagship species and surrogacy have been investigated before, the hypothesis that flagships are good surrogates has not been addressed directly before.

The general methodology used for the analysis is clear and indeed very well established in the literature. Yet, several decisions made by the authors in this analysis are unconvincing, and some of them in particular may have an influence on the results and conclusions. When testing the surrogacy potential of flagship species, one can expect that two parameters influence the result: the proportion of flagship species in the analysis, and the spatial constraints of possible solutions. The proportion of flagship species is likely to be non-linearly related to their surrogacy potential, and a saturation effect can be expected. One flagship species won't surrogate all other species very well. The first surrogate species added will add much surrogacy potential, then progressively adding other flagships will have a diminishing return in terms of surrogacy potential. So the proportion of flagship species chosen for the analysis is crucial.

1. A few decisions seem to inflate the number of flagship species used:

- it is not clear to me why Near Threatened species were included. They are not considered threatened with extinction by the IUCN, therefore they are unlikely to get conservation attention due to their level of extinction risk

- it is unclear what the "original conservation flagships" mentioned in the Methods refers to. Anyway, adding cinderella species means inflating the number of species considered flagship in the analysis. While the idea of cinderella species is intriguing, in practice they are not used for fundraising

- similarly, the addition of "The top 100 reptile and 500 bird species, measured by total [web] page views" should be better justified. What is the actual count of page views? Do all top 100 reptiles have a chance to be true flagships? Or the birds that are not at the top of the list?

2. Other decisions seem to constrain solutions in space in a way that may force flagship species to perform well:

- the whole analysis is limited to the Global 200 ecoregions, which are already a very special subset of the world in terms of importance for biodiversity. I wonder how many planning units did not contain at least one flagship species

- Most flagship species are large mammals that typically need large extents of undisturbed habitat. I suspect that constraining the analysis to areas with low Human Footprint further reduces options for the prioritization analysis towards places that contain flagship species

- I did not find the choice of the random test convincing. It seems that the random test just selects places at random. But what would be the surrogacy of a random sample of 534 species (the number of flagship species eventually used in the analysis)?

3. Additional minor point: I think that the parts of the species ranges that are flagged by IUCN as "possibly extinct" should not be included in the analysis to avoid overestimating distributions.

Overall, I find the ms potentially useful but I don't think that the analysis in this form can answer the question properly. I would like to see a test of the effect of proportion of flagship species, and a more careful randomization test.

Reviewer #2 (Remarks to the Author):

McGowan et al present an analysis to understand, as they note, whether the best way of raising money (specifically through the use of flagship species fundraising campaigns) excludes the best way of spending it (on species in the greatest need of conservation). This is an excellent question to test and an important field to undertake empirical research on, although I have a few concerns with the approach and the way the results are presented by authors.

I appreciate that this manuscript is primarily about place-based conservation, and specifically on the understanding that protecting habitat for places where charismatic species occur will thereby also lead to collateral benefits for other species that co-occur. That's true to a point but my major issue is that, while critical, habitat-based conservation isn't always the required conservation need (I realize the authors recognize this: lines 2016-212). Hence, funds raised to do anti-poaching patrols and to implement SMART to deliver benefits for tiger conservation, won't impart any benefit to a co-occurring amphibian species impacted by disease or an invasive species, or a co-occurring freshwater species affected by aquatic pollution. So my first point is that I think there needs to be fuller recognition (ideally in the abstract) that even if flagship species (in the broader sense as used in this paper) do a bang-up job of capturing places where other threatened species occur, the actions that are put in place to conserve those flagship species won't necessarily benefit co-occurring background species.

How well the flagship species model performs depends on what we call a flagship species in the first place. The current manuscript rightly uses flagship species in a broader sense of the word (*sensu lato* for want of a better term). While many organizations (lines 53-55) are selecting a wider suite of flagship species, the issue is that flagship species tend to still be used in a narrow sense (*sensu stricto*) and equated with tigers and elephants – in fact, the authors themselves cite these as examples in the first line of the abstract of the paper... A gander at USFWS species funding opportunities includes species that one might still traditionally consider flagships in the narrow sense. Hence, the results of this paper strongly hinge on marketing departments in conservation organizations and donors adopting flagships *sensu lato* and in that case I'm sure that the assertion that "prudently selected flagships can both raise funds for conservation and help target where these resources are best spent for broader biodiversity" will prove to be true. But if it only leads to support for more funding for tigers and elephant and pandas by donors and NGO's pushing this agenda even harder, then broader biodiversity will lose out. My second point, then, is that I think this needs to be acknowledged. In fact, I would urge the authors to consider including an analysis that compares the performance of traditional flagships (*sensu stricto*; e.g. one could simply exclude the Cinderella species) with the broader interpretation of flagships as that may help illustrate the difference in terms of performance.

My final point is really that, ultimately, how well flagship species perform (noting that many, but not all, flagships tend to be threatened) should be about how well they do at capturing threatened background biodiversity. This is not really addressed in the paper and I think this is a key shortcoming. I admit to not understanding why the authors selected to use the approach they did (which is somewhat removed from real-world application) – given the place-based focus of their analysis, why not also use a dataset with known species occurrences and test how well flagships do at capturing threatened background species? If you really want to test whether the best way of raising funds excludes the best way of spending it, then start with the best way of spending it. For a start, how well do flagships perform at capturing sites that 100 different conservation organizations all agree are truly globally important for conservation (lines 83-84), like Alliance for Zero Extinction sites? This is the real test – do flagship species capture critical sites like the unprotected Massif de la Hotte with its 10+ threatened amphibian species and in need of urgent habitat protection? This is far, far more meaningful for actual conservation on the ground.

Just a few minor points:

Lines 23-24: I'm not sure this really summarizes the major criticism? To my mind, the major criticism is that: i) too much money goes to flagships and not enough to the myriad of other species in urgent conservation need (this isn't the focus of this ms, but the authors do reference it in lines 52-53); and ii) places important for tigers, pandas and elephants (flagships sensu stricto and possibly lato) don't always capture other threatened biodiversity.

Lines 29-30: as noted above, I think this objective should be focused on threatened species

Author Response to Reviews

We thank the editors and reviewers for the chance to improve and resubmit our manuscript and for your encouraging comments on the value of this work.

Below we synthesise and respond to the primary concerns emerging from the review process. We follow this discussion with a point-by-point response to the more specific comments raised by each reviewer.

Our analysis is not a test of whether flagship species are good biodiversity surrogates

Both reviewers mention or imply that our work seeks to test the surrogacy value of flagship species. This was not the aim of our work but we recognise that by basing our measure of effectiveness on background species conserved, we made it easy to confuse this with surrogacy. To resolve this, we have put serious effort into making this issue clearer throughout the manuscript, removing any mention of surrogacy in the manuscript and adding text to explain the difference between our analysis and the previous work that looked at surrogacy. We have also revised Fig S1, the schematic of the algorithm. To explain this issue more, we also thought it would be helpful to provide background detail here:

Much of the earlier work on flagship species assumed that flagships were a biological surrogate species concept, similar to keystone, umbrella, or indicator species. This has changed in recent years with the recognition that flagship species are actually a marketing tool, as outlined in the paper by Verissimo et al (2011) which defined them as “a species used as the focus of a broader conservation marketing campaign based on its possession of one or more traits that appeal to the target audience”. This means that the effectiveness of a flagship species always depends on the campaign objectives and the target audience.

Our analysis was based on a real-world scenario, where a NGO (a) wants to identify the most important places for conservation and (b) fundraise to conserve those places based on the presence of species that will appeal to their target audience of international donors. By needing each place to contain charismatic species, the NGO has added an additional constraint when selecting priority places. Our analysis tested whether adding this constraint reduced the total number of species represented in the selected places, which is an entirely new question for the field. We would argue our analysis is the correct approach to understanding whether a flagship-based approach compromises conservation goals when selecting priority places.

This is a different question to whether fundraising to conserve well-known charismatic species will lead to broader biodiversity also being conserved. Research has shown that this is often not the case, as many of these species are poor biodiversity surrogates. This is why recent papers have called for campaigns based on a wider range of flagship species (while recognising that many people love elephants, tigers etc and will not completely alter their donation patterns based on recommendations from the scientific literature).

What makes a suitable flagship species?

Both reviewers raised concerns regarding the selection and number of flagship species considered in our analysis. We feel these concerns partly reflect the reviewers' beliefs that we are testing the surrogacy potential of flagships to represent broader biodiversity, but this is not the case as we explain above. To make it clearer that flagship species are a marketing tool

rather than a biological concept, we have added the Verissimo et al (2011) definition to the first lines of our introduction.

The previous articles on the surrogacy values of flagship species also tended to look at a small number of species, focusing on the most famous ones. However, research on the number of flagship species used by conservation NGOs shows a much broader range. For example, Smith et al (2012) focused on threatened mammals found outside of developed countries and found NGOs were using 80 species and Verissimo et al (2017) found one NGO was using 97 species, including many non-threatened species (and a recent check shows they are now using 143 flagships). These articles also showed that there are many other species that could be used successfully as flagships with sufficient marketing effort.

This provides the background to our selection of flagships, which is based on the context of a NGO selecting a number of important places and then fundraising based on appealing to potential international donors. Thus, we selected species that would appeal to the target audience for this place-based campaign, rather than restricting our analysis to the dozen or so species that are most often used in species-based campaigns. To make this clearer, we have added text to the beginning of the Discussion explaining our flagship selection approach.

Lines 201-211: Our analysis departs from previous research focused on the ability of flagships to represent broader biodiversity (10, 11) by looking at an entirely new research question, namely, whether important places for biodiversity contain flagship species. Our analysis also widened the number of flagship species than those used in previous studies, which often focused on the most famous charismatic megafauna. Increasing the number of flagships obviously reduces the constraint imposed. However, all of the species we used are either already popular with the target audience of potential donors in higher income countries, or are similar in size and appearance to these species. This means the range of flagship species in our study better reflects those already adopted by conservation NGOs.

Are the objectives we set the right objectives?

The novel advance of this paper is to provide an integrated problem-based approach (e.g. maximising or minimizing an objective, or objectives) to identify places for conservation *and* species that can be used to raise money for the conservation of those places. Objectives should be seen as the quantitative translation of values.

To illustrate our approach, we designed our objective to maximize the number of background species. We felt that this was a realistic conservation objective as it represents all species and important ecoregions in line with Aichi target 11 and 12. Our analysis tests the performance of our integrated approach for this objective given a variety of place-based constraints around protected areas, wilderness and ecoregions. We acknowledge this objective and these constraints reflect a set of values.

Reviewer 2 preferences a different set of values evidenced by the comments: “*flagship species perform[ance] should be about how well they do at capturing threatened background biodiversity*” and “*how well do flagships perform at capturing sites that 100 different conservation organizations all agree are truly globally important for conservation (lines 83-84), like Alliance for Zero Extinction sites?*”

We agree these are important and as such, could be translated into an objective that maximizes the number of threatened background species and/or uses the place-based

constraint of including sites for zero extinction. Ours, and Reviewer 2's, are examples of many possible problem-constructions to which our integrated approach could be applied.

We chose a range of plausible values (e.g. G200 ecoregions, low human footprint and Protected Areas), but could just as easily apply the method to any of the conservation templates provided in Brooks et al. (2006) such as: frontier forests, crisis ecoregions, centres for plant diversity, megadiverse countries, etc). We have tried to make this important message clearer in the Discussion and highlight the perspective of Reviewer 2:

Lines 223-230: When choosing places, organizations invariably have different perspectives on the most important constraints and enabling factors to consider relative to their conservation values. There are a number of spatially mapped biodiversity assets that could serve to inform the selection of candidate places (e.g. hotspots of species richness, remaining wilderness, Alliance for Zero Extinction sites (zeroextinction.org), Key Biodiversity Areas (keybiodiversityareas.org), climate refugia, etc (24) and biodiversity benefits (e.g. ecosystem services, threatened species, or phylogenetic diversity (16).

See the point by point response below for particular in-line changes to the manuscript. We reiterate that we are advocating for the *method* we present, rather than any specific set of flagships or priority locations in the Discussion.

Providing a different null model

The reviewers both encouraged a second null model to supplement the analysis and we agree, there are likely several very interesting additional randomizations we could test.

We designed the original place-based null model to select a random place from the background species matrix and see how well the benefit could be captured for the same number of sites identified by our integrated approach across scenarios without the additional constraints of flagship species or ecoregional complementarity. Given our objective is not about testing surrogacy, we feel it appropriate to keep the place-based null model as the primary randomization in the main text, as it was designed to follow the same logic of the place-based and integrated approaches we present and compare.

However, based on the reviewers' suggestion, we have conducted an additional randomized test. We designed the new test to identify random flagship species from the candidate list, then identify the location in the randomly selected species range that provides the largest benefit (e.g. maximum number of background species). The algorithm then follows this selection until all background species have been represented. We compare the efficiency of this approach as we do all others - by comparing the benefit delivered for the number of sites found in the integrated approach. As expected, we found this approach performs reasonably well as it still aims to maximise the background species benefit, in some instances performing better than our integrated approach because the constraint of ecoregional representation is not imposed on the null models. Interestingly, these results do not hold when we evaluate the top 10 most efficient sites, and performance efficiency of this species-based null model spanned between 72-74% compared to our integrated approach 87-97%.

This was a useful exercise and we thank the reviewers for suggesting this additional test. We provide the results of this test in the same format as Figure 1- which compares the accumulation curves of the 100 null tests for each scenario against the integrated, and place-only approaches. The additional species-based null model results are provided in the Supplementary materials, which include box plots (S3b), a table of comparative results (Table S2) and the accumulation curves comparison (Fig S4).

More information is available in the point by point response to Reviewer 1 below.

Reviewers' comments:

Reviewer #1 (Remarks to the Author):

This ms aims to assess whether flagship species, which are useful for attracting conservation funds, are also good surrogates for other species when identifying priority areas for conservation in a conservation planning framework. The question is important, because if flagships are also good surrogates, then the money spent for their conservation also benefits other species threatened with extinction. Although several questions around flagship species and surrogacy have been investigated before, the hypothesis that flagships are good surrogates has not been addressed directly before.

The general methodology used for the analysis is clear and indeed very well established in the literature. Yet, several decisions made by the authors in this analysis are unconvincing, and some of them in particular may have an influence on the results and conclusions. When testing the surrogacy potential of flagship species, one can expect that two parameters influence the result: the proportion of flagship species in the analysis, and the spatial constraints of possible solutions. The proportion of flagship species is likely to be non-linearly related to their surrogacy potential, and a saturation effect can be expected. One flagship species won't surrogate all other species very well. The first surrogate species added will add much surrogacy potential, then progressively adding other flagships will have a diminishing return in terms of surrogacy potential. So, the proportion of flagship species chosen for the analysis is crucial.

1. A few decisions seem to inflate the number of flagship species used:

- it is not clear to me why Near Threatened species were included. They are not considered threatened with extinction by the IUCN, therefore they are unlikely to get conservation attention due to their level of extinction risk.

To test the sensitivity of our results to a smaller list of candidate flagships, we used the IUCN threat status as an attribute for subsetting. We chose to use IUCN threat status because it is consistent across taxa and, as the reviewer suggests, relevant to conservation fundraising potential. We used Near Threatened as a proactive threshold for species in need of action in order to prevent further risk of extinction in the near future. This criteria subset our full candidate list 534 to 338 and Table 2 in the main document shows the number of potential candidates considered for each scenario, which ranges from 534-207.

We point out here that extinction risk is often not the driver for many existing flagship campaigns which instead often promote species of Least Concern and Near Threatened status. WWF US, for example, who are large proponents of using conservation flagships (found at: <https://gifts.worldwildlife.org/gift-center/gifts/Species-Adoptions.aspx>) promote jaguar (Near Threatened); Grey wolf (Least Concern), Red-tailed Hawk (Least concern), and Eastern grey Kangaroo (Least Concern) among many others.

Regardless, we agree there was inconsistency in the MS around how we frame threatened status. We have changed the "Threatened Flagship" title in Fig 1 to "Threat Status" so as to not suggest that Near Threatened is equivalent to currently Threatened species and added the NT-EN status range of status considered for this candidate grouping in the legend.

We have also added more descriptive information on the proportion of candidate flagships falling into different threat classes into the manuscript and added the following table to the Supplementary material as Table S1b.

Lines 124-129: “These species were then classified according to their IUCN Red-List status (www.iucnredlist.org) of which 37% were Least Concern; 6% were Near Threatened; 21% were Vulnerable; 21% were Endangered, and 15% were Critically Endangered; see Table S1b).”

IUCN_status	Candidate Bird	Candidate Mammal	Candidate Reptile	Grand Total
Critically Endangered	12	62	4	78
Endangered	12	98		110
Vulnerable	16	90	7	113
Near Threatened	27	7	1	35
Least Concern	160	4	27	191
Lower Risk/conservation dependent			1	1
Lower Risk/least concern			5	5
Lower Risk/near threatened			1	1
Grand Total	227	261	46	534

This comment was also helpful in highlighting that because multiple flagships often occur in many of the places that emerge for a scenario (up to 20 in some places in scenario h);), we should mention that additional criteria could help select which species ultimately become flagships by the end user. We added the following to the manuscript:

190-192: “Multiple flagships found in an individual place provides flexible options for organizations to select flagship species that best reflect their conservation strategies and donor preferences, while also representing these globally unique ecoregions.”

- it is unclear what the "original conservation flagships" mentioned in the Methods refers to. Anyway, adding cinderella species means inflating the number of species considered flagship in the analysis. While the idea of cinderella species is intriguing, in practice they are not used for fundraising.

Similarly, the addition of "The top 100 reptile and 500 bird species, measured by total [web] page views " should be better justified. What is the actual count of page views? Do all top 100 reptiles have a chance to be true flagships? Or the birds that are not at the top of the list?

This comment speaks to an important assumption we make in our candidate flagship list that we had not made clear in the manuscript. We hope that our explanations of conservation flagships and the general discussion in our response is enough to suggest that the flagship species we selected would be appropriate for the type of place-based flagship campaign we describe above. We have added the following assumption to the manuscript in

Lines 277-279: “We assumed that all species in the list of candidates have equal capacity to serve as a conservation flagship given dedicated marketing efforts.”

We have removed “original” flagships from the text in the Methods. This statement refers to the existing flagship mammals identified in Smith et al. (2012) which formed the basis of the statistical analysis conducted to identify the Cinderella species.

There is evidence that Cinderella species have been used as flagships, for example, the Tamaraw, which lives in the Philippines. <https://www.globalwildlife.org/our-work/regions/asia/conserving-the-natural-and-cultural-heritage-of-mounts-iglit-baco/>

2. Other decisions seem to constrain solutions in space in a way that may force flagship species to perform well:

- the whole analysis is limited to the Global 200 ecoregions, which are already a very special subset of the world in terms of importance for biodiversity. I wonder how many planning units did not contain at least one flagship species.

- Most flagship species are large mammals that typically need large extents of undisturbed habitat. I suspect that constraining the analysis to areas with low Human Footprint further reduces options for the prioritization analysis towards places that contain flagship species.

To better clarify how the constraints on species and places influenced the prioritizations across the scenarios, we have significantly changed Table 2. The table now includes the total number of sites available and the total number of candidate flagships for each of the eight scenarios. We note that across the scenarios, the range of candidate species considered spans from 534 – 207 and that the inclusion of a candidate flagship species is a constraint we placed on the prioritization algorithm.

Below we provide the answer to the above question regarding how many planning units did not contain a flagship species across all scenarios. The influence of these sites on the prioritization can potentially be linked to the lost efficiency in our integrated approach if these sites also happen to have a large amount of background species. This relationship is captured in our comparison to the purely greedy “place-based” approach, which uses the same background species richness without the constraint of having to have a flagship species in the site.

These results show that most places contain at least one flagship species, especially when Least Concern flagships are included in the candidate set. Place-based constraints (in grey in the table below) had between 0 and 0.0008 of their locations without flagship species, so the Human Footprint constraint does not have more influence than other constraints when it comes to the resulting candidate places containing flagship species. However, our analysis was designed to inform a real-world place-based approach and, as we have argued above, we think that this would be based on a much wider range of flagship species than those used in studies of the surrogacy value of species-based campaigns.

Scenario	No. of sites without flagships	No. of sites	Prop without flagships
A. Ecoregions	8	10200	0.0008
B. Ecoregions + Threat Status	1867	10200	0.18

C. Ecoregions + Protected Areas	0	3097	0
D. Ecoregions + Protected Areas + Threat Status	464	3097	0.15
E. Ecoregions + Human Footprint	1	3961	0.00025
F. Ecoregions + Human Footprint + Threat Status	1097	3961	0.28
G. Ecoregions + Protected Areas + Human Footprint	0	1069	0
H. Ecoregions + Protected Areas + Human Footprint + Threat Status	267	1069	0.25

- I did not find the choice of the random test convincing. It seems that the random test just selects places at random. But what would be the surrogacy of a random sample of 534 species (the number of flagship species eventually used in the analysis)?

As stated above, we designed the original null model to follow the same logic of the place-based and integrated approaches we present and compare for the objective of our analysis. However, we have conducted a second randomized test based on selecting random species and we provide the additional results in the Supplementary materials in Table S2 and Figure S3b and Figure S4 (below). We feel it is very important to restate that because we were not testing the surrogacy potential of flagships, this null model is simply a different variation of our integrated approach which focused on species first, rather than places. This randomized test still aims to maximize the benefit at each step, and so without the additional constraint for representing ecoregions in the solutions, it unsurprisingly performs quite well. What is interesting is the poor performance of the first 10 sites compared to our integrated place-based approach. We keep this analysis in the Supplementary materials to not confuse the reader.

Results from the place-only, integrated, and random species approaches. Values describe the maximum benefit available for each scenario (Scen. a-h) as described in Table 2 and Fig S4. We report on the mean benefit of 100 prioritizations for the randomized tests.

Scenario	Max benefit (No. of background species)	No. Sites in solution:	Number of species delivered with:			Percent of Efficiency retained for full suite of sites:		Percent of Efficiency retained for top 10 sites:	
			place-only approach	integrated approach	Species-based random test	integrated	Species-based null (mean 100 runs)	integrated	Species-based random test
a	19,616	107	12,878	10,545	11,493	82%	89%	87%	72%
b	19,616	84	11,961	9,487	9,448	79%	79%	90%	68%
c	16,542	93	11,835	9,965	10,596	84%	90%	92%	74%
d	16,542	83	11,443	9,387	9,220	82%	81%	92%	69%
e	12,053	77	9,362	7,972	8,391	85%	90%	89%	74%
f	12,053	58	8,557	7,621	6,935	89%	81%	96%	71%
g	9,833	62	8,363	7,269	7,400	87%	88%	93%	74%
h	9,833	47	7,702	6,849	6,017	89%	78%	97%	71%

Fig S4. Performance comparisons across scenarios. The scenario performance of the place only, integrated, and random flagship species approaches (a-h) in achieving the conservation benefit (defined as the number of background species protected). Random selections were performed 100 times for each scenario. Threat Status refers to the second candidate flagship group driven by IUCN classification of Near Threatened and higher.

3. Additional minor point: I think that the parts of the species ranges that are

flagged by IUCN as "possibly extinct" should not be included in the analysis to avoid overestimating distributions.

We followed the treatment of IUCN species ranges laid out in the important paper by Butchart et al. (2015) Shortfalls and Solutions for Meeting National and Global Conservation Area Targets. *Conservation Letters* **8**, 329-337 (2015).

Because we are not testing surrogacy potential in this analysis, we feel the inclusion of these sites remains justified. We do not assume the conservation actions of the intended user or organization, which could include species reintroductions to these “possibly extinct” locations. We have added the following to the manuscript to clarify this assumption

Lines 236-238 “Conservation investments will always be tied to actions (e.g. securing specific parcels of land, restoring degraded habitat, tackling invasive species, anchoring an organization in a new landscape, lobbying a government, **species reintroductions**, etc.).”

Overall, I find the ms potentially useful but I don't think that the analysis in this form can answer the question properly. I would like to see a test of the effect of proportion of flagship species, and a more careful randomization test.

Reviewer #2 (Remarks to the Author):

McGowan et al present an analysis to understand, as they note, whether the best way of raising money (specifically through the use of flagship species fundraising campaigns) excludes the best way of spending it (on species in the greatest need of conservation). This is an excellent question to test and an important field to undertake empirical research on, although I have a few concerns with the approach and the way the results are presented by authors.

I appreciate that this manuscript is primarily about place-based conservation, and specifically on the understanding that protecting habitat for places where charismatic species occur will thereby also lead to collateral benefits for other species that co-occur. That's true to a point but my major issue is that, while critical, habitat-based conservation isn't always the required conservation need (I realize the authors recognize this: lines 2016-212). Hence, funds raised to do anti-poaching patrols and to implement SMART to deliver benefits for tiger conservation, won't impart any benefit to a co-occurring amphibian species impacted by disease or an invasive species, or a co-occurring freshwater species affected by aquatic pollution. So my first point is that I think there needs to be fuller recognition (ideally in the abstract) that even if flagship species (in the broader sense as used in this paper) do a bang-up job of capturing places where other threatened species occur, the actions that are put in place to conserve those flagship species won't necessarily benefit co-occurring background species.

We agree entirely. We are promoting place-based conservation to deliver benefits at the local scale. Notably, the broad issue of “protection” of a place means both the protection of the habitat in that place and removal of a wide range of common threats in that place. We acknowledge the full suite of actions that may need to be implemented in the priority places and added the following to the discussion:

*Lines 241-248: “Since the aim of this paper was to test the efficiency of our integrated approach for a range of global scale analyses, we did not prioritize for any particular action. **Instead, we would encourage conservation practitioners to evaluate which of the available conservation actions are most suitable in each of the priority places to benefit broader biodiversity.** We suggest our approach be considered complementary to more specific systematic conservation planning activities and decision theory approaches, which can further optimize the most appropriate placement and timing of management actions at finer scales.”*

We have also completely rewritten the abstract from the first submission in order to meet the 150 word limit.

How well the flagship species model performs depends on what we call a flagship species in the first place. The current manuscript rightly uses flagship species in a broader sense of the word (sensu lato for want of a better term). While many organizations (lines 53-55) are selecting a wider suite of flagship species, the issue is that flagship species tend to still be used in a narrow sense (sensu stricto) and equated with tigers and elephants – in fact, the authors themselves cite these as examples in the first line of the abstract of the paper... A gander at USFWS species funding opportunities includes species that one might still traditionally consider flagships in the narrow sense. Hence, the results of this paper strongly hinge on marketing departments in conservation organizations and donors adopting flagships sensu lato and in that case I’m sure that the assertion that “prudently selected flagships can both raise funds for conservation and help target where these resources are best spent for broader biodiversity” will prove to be true.

But if it only leads to support for more funding for tigers and elephant and pandas by donors and NGO’s pushing this agenda even harder, then broader biodiversity will lose out. My second point, then, is that I think this needs to be acknowledged. In fact, I would urge the authors to consider including an analysis that compares the performance of traditional flagships (sensu stricto; e.g. one could simply exclude the Cinderella species) with the broader interpretation of flagships as that may help illustrate the difference in terms of performance.

We hope our clarification in the beginning of this response about what makes a good flagship helps address this concern. We also added a discussion of the expanded list of potential flagships in the discussion:

Lines 204-210: “Our analysis also expands the number of flagship species than those used in previous studies, which often focused on the most famous charismatic megafauna. Increasing the number of flagships obviously reduces the constraint imposed. However, all of the species we used are either already popular with the target audience of potential donors in higher income countries, or are similar in size and appearance to these species. This means the range of flagship species in our study better reflects those already adopted by conservation NGOs.”

My final point is really that, ultimately, how well flagship species perform (noting that many, but not all, flagships tend to be threatened) should be about how well they do at capturing threatened background biodiversity. This is not really addressed in the paper and I think this is a key shortcoming. I admit to not understanding why the authors selected to use the approach they did (which is somewhat removed from real-world application) – given the place-based focus of their analysis, why not also use a dataset with known species occurrences and test

how well flagships do at capturing threatened background species? If you really want to test whether the best way of raising funds excludes the best way of spending it, then start with the best way of spending it. For a start, how well do flagships perform at capturing sites that 100 different conservation organizations all agree are truly globally important for conservation (lines 83-84), like Alliance for Zero Extinction sites?

This is the real test – do flagship species capture critical sites like the unprotected Massif de la Hotte with its 10+ threatened amphibian species and in need of urgent habitat protection? This is far, far more meaningful for actual conservation on the ground.

We hope the discussion of the question we are asking and the objectives in the synthesis of our response sufficiently speaks to this point. Many conservation groups are interested in protecting a portfolio of sites (using flagships to market those sites) where that portfolio of sites delivers representative conservation. Such organisations include The Nature Conservancy and the Wildlife Conservation Society where representativeness and comprehensiveness remain core conservation goals. The reviewer is correct when suggesting that additional preferences may also inform the problem construction and so we have changed the manuscript to highlight AZE sites when we discuss alternate place-based constraints:

Lines 223-233: “When choosing places, organizations invariably have different perspectives on the most important constraints and enabling factors to consider relative to their conservation values. There are a number of spatially mapped biodiversity assets that could serve to inform the selection of candidate places (e.g. hotspots of species richness, remaining wilderness, Alliance for Zero Extinction sites (zeroextinction.org), Key Biodiversity Areas (keybiodiversityareas.org), climate refugia, etc (24) and biodiversity benefits (e.g. ecosystem services, threatened species, or phylogenetic diversity (16). Our approach allows organizations to tailor the problem definition to their objectives, moving beyond static asset maps and towards identifying priority places for conservation action by considering them within a properly constructed problem.”

Just a few minor points:

Lines 23-24: I’m not sure this really summarizes the major criticism? To my mind, the major criticism is that: i) too much money goes to flagships and not enough to the myriad of other species in urgent conservation need (this isn’t the focus of this ms, but the authors do reference it in lines 52-53); and ii) places important for tigers, pandas and elephants (flagships sensu stricto and possibly lato) don’t always capture other threatened biodiversity.

Lines 29-30: as noted above, I think this objective should be focused on threatened species. We agree these are important points but threatened species were not the objective we defined in our analysis- as we focused on the representation of all biodiversity. We hope that our discussion of the many types of values our integrated approach can incorporate speaks to the flexibility of our method for users to define different objectives, of which threatened species could be one.

Reviewers' comments:

Reviewer #3 (Remarks to the Author):

I had the chance to read the revised ms together with the comments provided by a first round of review and the relative responses provided by the authors. Therefore, I tried to focus on the most important issues considering the previously highlighted ones, and added some points that I found of particular importance.

I should admit I have really contrasting feelings when reading the article. On the one side, I acknowledge the importance of the topic, appreciate the global perspective provided on the subject, enjoy the different scenarios explored. I also found the paper to be -in general, but with partial exceptions- well written. Therefore, I find it as an interesting and well conducted exercise. The other side of the coin is that, as far as I can see, this is exactly an exercise and no more than this, and, additionally, from my point of view this is an exercise for which it is rather hard to envision possible applications in the real world.

I guess that, if it has to be demonstrated that “flagship species can deliver efficient conservation”, some ‘real’ conservation results should be provided, and definitely much more than a very broad-scale prioritization exercise is required.

In particular, in addition to some concerns already highlighted by the previous reviewers, I find the following points particularly concerning:

1. could be “user-inspired” (l. 92) an approach based on 100x100 km grid squares? Decision makers, stakeholders, NGOs and other conservation agencies mostly work at much finer scales;
2. what is the ecological/biological meaning of working at such a coarse level? As the authors acknowledge (ll. 291-292), the planning unit size is very coarse compared to many species ranges;
3. is species presence/absence at such a broad scale really meaningful for conservation? Weighting the same the occurrence of one individual or of 100,000 within a 100x100 km square is not correct when quantifying the importance for conservation. I acknowledge that having an estimate of species richness is probably one of the best measures one could obtain at a global level while including thousands of species, but I am worried that this is indeed useful for an academic exercise, not for real conservation.

Considering developed countries, or especially mountain regions, the environmental heterogeneity and the landscape/habitat fragmentation often imply that, within a 100x100 km square, there are a lot of different environmental contexts, which can not be ‘captured’ by using one or a few flagship species, and which likely require different conservation strategies/interventions, that are not necessarily related in any way with flagship species. The potential importance of flagship species in attracting funds actually useful for conservation thus is hard to be properly evaluated at such a scale.

These are the reasons why I am not sure that the analysis can answer “a question of real-world relevance for conservation practitioners”; rather, it seems to me that this work is a sort of a finer

assessment of surrogacy value (despite the authors' claims in the response to reviewers) intersected with other factors potentially conditioning both conservation values and conservation constraints. Therefore, I think that the promise included in the title is not kept in the development of the paper. What is actually offered is that the large-scale selection of macroareas for conservation at the global level could usefully integrate the use of flagship species (to increase funding opportunities), or that the careful use of the latter may increase the efficiency of macroarea selection for conservation within priority regions. Following the ideas (adopted also by the authors in the text) that i) flagship species may attract funds, ii) species can become flagship with adequate marketing, iii) flagship species should be used to provide funds for important areas, paradoxically I would see more logical to identify the 'right flagship species' for each important area (based not only on occurrence, but on actual association with e.g. broader biodiversity and ecosystem functions – carefully evaluated surrogacy value), and promote marketing on it...

In conclusion, I suggest to re-frame the paper considering these potential weaknesses.

Specific comments

II. 212-213: I think that real-world conservation practitioners need much finer scales and much more detailed data than presence/absence over 100x100 km cells.

II. 235-247: this sounds a bit like: "we have proposed a method to identify some areas, but conservation in practice should be based on completely different stuffs". I acknowledge that this was probably added in response to a previous reviewer's comment, but it further strengthens the impression of an exercise poorly linked with real-world questions. Assuming to use flagship species to attract funding for conservation actions, it could be expected that such actions should be somewhat related to flagship species, but as the reviewers already noted, this actually does not imply benefits for other species, imperiled species or broader biodiversity.

II. 275-277: "We assumed that all species in the list of candidates have equal capacity to serve as a conservation flagship given dedicated marketing efforts" I acknowledge that some simplifications are needed, but this is very unlikely – marketing efforts can not always overcome people perceptions, due to e.g. cultural background or economic conflicts (even if more perceived than real), as the stories about many large, charismatic, carnivores in developed countries perfectly exemplify.

Reviewer #4 (Remarks to the Author):

This paper presents a novel analytical approach that could make an important contribution to the field of conservation prioritization. While I did not review previous versions, it seems that the authors were diligent and thorough in addressing prior comments. The authors demonstrate a high degree of spatial overlap in terrestrial regions with high overall biodiversity and those that contain charismatic flagship species commonly used in conservation marketing and fundraising campaigns by

NGOs. Their prioritization approach shows that prioritizing areas for conservation based on the presence of flagship species results in the selection of areas with ~80-90% of the total species that would be selected by maximizing only the total number of species. Further, this approach outperforms a random selection of areas in terms of the number of species occurring in selected areas. Their results provide an interesting and useful bridge across the juxtaposition between conservation targeting single, charismatic species (which are often successful fundraisers) and biodiversity conservation.

I found the manuscript at times went beyond what the results of the novel prioritization approach demonstrates. Some of this stems from the writing style and could be addressed with more plain and explicit language. In my opinion terms like 'conservation benefit' and 'biodiversity objective' can obfuscate what was ultimately analyzed in the paper – the number of species occurring in areas of potential conservation priority that also contain flagship species. More precisely describing these metrics throughout would not only make the manuscript read more easily, but make the methodology more easily understood and convey what the analyses discovered.

Closely related to this point, conceptually I think it is important to acknowledge and distinguish the difference between overlap in coarse (100 km²) species presence, and shared characteristics between flagship and background species that would allow the latter to benefit from funding conservation actions directed at the former. The authors should identify and clarify the many intermediate processes that might cause a disconnect between the spatial overlap of species and their ability to mutually benefit from conservation actions (e.g. differences in habitat, shared threats, range overlap, legal protections, etc). The authors are clear in acknowledging that funding raised in the service of a single species and/or its habitat cannot be ethically spent on unrelated conservation programs. However, much of the framing and discussion seems to make an implicit assumption that money being directed to an area provides conservation benefit to the species in that area by default. This is related to the use of broad terms such as those identified above.

More specific comments are provided below.

Line 23: This framing, to me, is not quite accurate. The criticism of flagship species is not based on where money goes, but that it is only directed towards single species. I think plenty of conservationists would acknowledge that funding for flagship species is spent in biodiversity rich areas. The criticism is based on what actions those resources fund and whether those ameliorate threats for one or many species. This may seem like a minor nuance, but the distinction is important to clarify as it underpins the broader framing of this paper.

Line 27: "Biodiversity objective" is a bit obscure here before any context is given. It might be clearer, if not more accurate, to say something like that you identified places that contain flagship species and maximize biodiversity representation.

Line 66: “Here we ask to what extent is fundraising through a flagship species approach a major constraint on delivering efficient place-based conservation at a global scale?” Consider rephrasing this question to more closely align with the outcomes of the optimization analyses. The phrasing at Line 204 would be clearer.

Line 106: I appreciate the desire to frame and discuss this work in a rigorous objective-maximization framework, but I think clarity for the reader could be gained by stating more explicitly the measure you evaluated (i.e. number of background species encompassed) rather than “80 – 89% of the objective.”

Line 115 – 141: These paragraphs read primarily like methods to me, laying out the steps used to generate different subsets of species and areas to be analyzed. Consider relocating the section to the Methods. At the very least, the subsection title should be changed from ‘the influence of preferential attributes on candidate flagship species and places’ to something more straightforward like ‘flagship species and focal area selection.’

Line 145: “performance efficiency?” Unclear what this refers to, as the optimization approach hasn’t been fully described. It would also seem like one would either measure the performance or the efficiency of the approach, but not ‘performance efficiency?’ I may be misunderstanding if this term is a precise reference to a more specific metric, and if so it should be defined before it appears here.

Line 148 – 150: While you are clear in defining ‘conservation benefit’ as the number of background species present in a given area, I would prefer a more specific term because of the different factors that would contribute to what most readers would call conservation benefit (i.e. species protections) that can’t be accounted for in the framework addressed in this paper (e.g. habitat/niche overlap between flagship species and background species, commonality of threats, legal authority, etc.)

Line 176: Again, its not clear how you’re measuring efficiency, or what this refers to. Is this area under the species area curves?

Line 204: This strikes me as the clearest, most accurate description of the research question addressed – “whether important places for biodiversity contain flagship species.” I recommend incorporating this phrase, or similar descriptions, when describing the study in preceding sections.

Line 235 – 247: This is an important point to acknowledge, that it is conservation actions which ultimately deliver protections for species and conservation benefits. Whether in this paragraph or an additional Discussion paragraph, I think it is important to take this acknowledgement a step further and discuss how the degree of overlap between the available actions that benefit a flagship species and those that benefit background species. Although the degree of overlap between flagship species and background species was measured as presence-absence within 100km² cells, funds raised for

flagships will most often be limited to those species range or native habitat. These may have even less overlap with the ranges and habitats of background species. Some habitats are mutually exclusive, in which case there may be no overlap. If flagship and background species are not threatened by a common set of factors, even spatially overlapping species may not share in the conservation benefits of actions tailored to address threats for flagship species.

In the second round of revisions, we received two additional thoughtful reviews. We believe these reviews were fair and raised some important points to improve the framing and language of the manuscript. We have revised the text throughout the manuscript to address these concerns, with the majority of changes found in the Discussion.

Below, we synthesize the major changes we have made in response to these reviews, followed by a point-by point response to smaller criticisms and concerns.

1. Title and framing

We agree that the previous title “Flagship species can deliver efficient conservation” falsely raises expectations from the readership that the paper is about evaluating the efficiency of flagship species to deliver on-the ground conservation efforts. In reality, our paper is a different topic, presenting an approach to select flagships in order to prioritise places for conservation actions and testing the efficiency of that approach. In an effort to clarify this point, we have changed the title of our manuscript to better reflect our research.

The new title is “Resolving the flagship paradox: charismatic species do not compromise efficiency in conservation prioritization”

We believe this title better reflects the aim and findings of our paper- which is to encourage organisations who would like to use a flagship species approach, to do so systematically rather than using an ad-hoc approach, or based on charisma alone.

2. Real-world applicability

The reviews questioned the real-world applicability of this research. While we do not explicitly state this in the manuscript, we developed this approach to identify flagship species in direct response to a query from an environmental NGO. As a result, this approach is now being used to guide their global investment strategy (www.wildark.com) and we have also received considerable interest from three other NGOs interested in applying this approach to help guide their global protection and acquisition strategies. We also place a greater emphasis in the revised manuscript on the flexibility of our approach to accommodate different organizational values and objectives, and the ability of the methodology to tackle related problems.

3. Coarse scales and range maps to inform actions on the ground

At the global scale, our 100 x 100km planning grid size is not intended to be ecologically meaningful for local conservation actions but rather to inform upon the value of broad landscapes for conservation investment. We feel this concern was partially a response to the previous title and framing of the manuscript, which we have addressed (issue 1 above). We have differentiated between the areas we identify through a global exercise and the next steps to ensure decision-making can happen at the local scale - e.g. using our grid to identify a single locale which would serve as a focus for regionally relevant conservation. Further, we agree it is important to be clear about scale and resolution. Accordingly, we have added a section to the methods stating that common errors of omission and commission may occur when using the IUCN range maps. However, we also point to the findings of Marechaux et al. (2016) who examined the utility of coarse range maps to capture general patterns of biodiversity. These authors found that at global scales, coarse range maps provide good

estimates for informing priorities. We provide more details in the point-by-point responses below.

I. Maréchaux, A. S. L. Rodrigues, A. Charpentier. The value of coarse species range maps to inform local biodiversity conservation in a global context. *Ecography* **40**, 1166-1176. (2016).

4. The overlap of flagship species ranges with important biodiversity

Both reviewers raised concerns about the resolution of our analysis likely overestimating the direct overlap of background species ranges with the ranges of the flagships found in each cell. In our introduction, we highlight that the flagship species approach to conservation typically funds two types of projects: those that target and benefit the species directly (the approach that has caused much of the criticism of this approach because these species do not always act as good surrogates for biodiversity), and those that focus on broader issues, such as protecting or managing the land and seascapes where the flagship is found. Our analysis inherently speaks to the latter. In this light, it is less important if the range of a flagship species found in a grid square does not overlap with all the ecosystems and other species also found within it. For example, if an organization wants to raise money for protected area management in a coarse-scale grid square, the money raised can benefit freshwater biodiversity even if the funding campaign used a tiger in their marketing materials. We have made a few specific changes to articulate this point in the introduction and discussion and in lines 234-236, we stress that the actions to be funded should be made explicit at the beginning of a prioritization process.

Specific responses and revisions

Reviewer #3 (Remarks to the Author):

I had the chance to read the revised ms together with the comments provided by a first round of review and the relative responses provided by the authors. Therefore, I tried to focus on the most important issues considering the previously highlighted ones, and added some points that I found of particular importance.

I should admit I have really contrasting feelings when reading the article. On the one side, I acknowledge the importance of the topic, appreciate the global perspective provided on the subject, enjoy the different scenarios explored. I also found the paper to be -in general, but with partial exceptions- well written. Therefore, I find it as an interesting and well conducted exercise. The other side of the coin is that, as far as I can see, this is exactly an exercise and no more than this, and, additionally, from my point of view this is an exercise for which it is rather hard to envision possible applications in the real world.

I guess that, if it has to be demonstrated that “flagship species can deliver efficient conservation”, some ‘real’ conservation results should be provided, and definitely much more than a very broad-scale prioritization exercise is required.

Response: Thank you. We agree and have revised the title and manuscript accordingly to not give the impression that this paper is about delivering efficient conservation on the ground, but instead is designed to inform the broad areas in which an organization should focus (in response to a direct request from an environmental NGO).

In particular, in addition to some concerns already highlighted by the previous reviewers, I find the following points particularly concerning:

could be “user-inspired” (l. 92) an approach based on 100x100 km grid squares? Decision makers, stakeholders, NGOs and other conservation agencies mostly work at much finer scales;

Response: As highlighted above, our analysis is directly informing the work of a conservation NGO. To improve the text, we have clarified the important role of finer scale assessments to influence actions on the ground in the revised paragraph of the discussion starting lines 231-243 with new additions *in italics*:

“Investments will always be tied to actions (e.g. securing specific parcels of land, restoring degraded habitat, tackling invasive species, establishing an organization in a new landscape, lobbying a government, reintroducing species, etc.). Intended actions should be identified at the beginning of the prioritization, as their associated costs, benefits and feasibility will dictate which places emerge as priorities (30) and which species may be best suited to act as flagships. Ultimately, it is the responsibility of the fundraising organization to be transparent about what their investments will deliver for local biodiversity. We suggest our approach be considered complementary to more specific systematic conservation planning activities and decision theory approaches, which can further identify the most appropriate placement and timing of management actions at finer scales for practitioner’s working on the ground in multi-species landscape-scale conservation.”

what is the ecological/biological meaning of working at such a coarse level? As the authors acknowledge (ll. 291-292), the planning unit size is very coarse compared to many species ranges;

Response: Please see point 3 above (on page 1).

Is species presence/absence at such a broad scale really meaningful for conservation? Weighting the same the occurrence of one individual or of 100,000 within a 100x100 km square is not correct when quantifying the importance for conservation. I acknowledge that having an estimate of species richness is probably one of the best measures one could obtain at a global level while including thousands of species, but I am worried that this is indeed useful for an academic exercise, not for real conservation. Considering developed countries, or especially mountain regions, the environmental heterogeneity and the landscape/habitat fragmentation often imply that, within a 100x100 km square, there are a lot of different environmental contexts, which can not be ‘captured’ by using one or a few flagship species, and which likely require different conservation strategies/interventions, that are not necessarily related in any way with flagship species. The potential importance of flagship species in attracting funds actually useful for conservation thus is hard to be properly evaluated at such a scale.

Response: Please see points 3 and 4 above. With respect to scale, the resolution is appropriate because we are looking at broad areas of interest across the globe to leverage investment rather than prioritizing actions on the ground. We hope that the revised framing makes the aim of our paper clearer.

We have also added the following to the methods section titled “Background species.”

Starting lines 290:

Given that many species occupy ranges much smaller than our 100 km x 100 km planning unit size, we erred on the side of caution and did not assign a minimum size threshold to reflect species’ presence. The coarse resolution of global species range maps means our analysis is subject to errors of omission and commission (49). However, previous research shows that IUCN range maps provide good estimates to inform biodiversity priorities at global scales, but should be combined, when possible, with local data before finer –scale conservation decisions are made (50).

We have added this important reference that outlines our broader argument in great detail: I. Maréchaux, A. S. Rodrigues, Anne Charpentier. The value of coarse species range maps to inform local biodiversity conservation in a global context. *Ecography* 40, 1166-1176. (2016).

These are the reasons why I am not sure that the analysis can answer “a question of real-world relevance for conservation practitioners”; rather, it seems to me that this work is a sort of a finer assessment of surrogacy value (despite the authors’ claims in the response to reviewers) intersected with other factors potentially conditioning both conservation values and conservation constraints. Therefore, I think that the promise included in the title is not kept in the development of the paper. What is actually offered is that the large-scale selection of macroareas for conservation at the global level could usefully integrate the use of flagship species (to increase funding opportunities), or that the careful use of the latter may increase the efficiency of macroarea selection for conservation within priority regions. Following the ideas (adopted also by the authors in the text) that i) flagship species may attract funds, ii) species can become flagship with adequate marketing, iii) flagship species should be used to provide funds for important areas.

Response: Please see point 2 above.

We have also changed “practitioners” to “organizations” to better reflect that we are advising broad organizational conservation strategies rather than fine-scale decisions.

Comment: paradoxically I would see more logical to identify the ‘right flagship species’ for each important area (based not only on occurrence, but on actual association with e.g. broader biodiversity and ecosystem functions – carefully evaluated surrogacy value), and promote marketing on it...

Response: We agree and have edited the discussion starting lines 210:

.... we illustrate how the selection of flagships can be systematic and objective-based, given a set of conservation goals, target audience and marketing strategy, rather than *ad hoc* or driven solely by perceived charisma. Organizations should then choose flagships that align with the local ecological, conservation and social context (25, 40). Importantly, the flagship and background species do not need to have overlapping ranges within the priority places, as previous successful fundraising campaigns have simply used flagships to provide examples of the important biodiversity found within a priority region (12).

As well as Line 234:

Intended actions should be identified at the beginning of the prioritization, as their associated costs, benefits and feasibility will dictate which places emerge as priorities (30) and which species may be best suited to act as flagships. Ultimately, it is the responsibility of the fundraising organization to be transparent about what their investments will deliver for local biodiversity.

In conclusion, I suggest to re-frame the paper considering these potential weaknesses.

Response: We have revised the manuscript accordingly.

Specific comments

ll. 212-213: I think that real-world conservation practitioners need much finer scales and much more detailed data than presence/absence over 100x100 km cells.

Response: Please see the responses above.

ll. 235-247: this sounds a bit like: “we have proposed a method to identify some areas, but conservation in practice should be based on completely different stuffs”. I acknowledge that this was probably added in response to a previous reviewer’s comment, but it further strengthens the impression of an exercise poorly linked with real-world questions. Assuming to use flagship species to attract funding for conservation actions, it could be expected that such actions should be somewhat related to flagship species, but as the reviewers already noted, this actually does not imply benefits for other species, imperilled species or broader biodiversity.

Response: We agree. The nature of any global scale prioritization is to guide attention and focus to important regions rather than dictate what actions should happen on the ground. We feel this is an important point and have tried to highlight that finer scale assessments will need to be conducted in order to deliver conservation outcomes for local biodiversity. We caution that ultimately, this is the responsibility of the fundraisers who are using flagship species.

ll. 275-277: “We assumed that all species in the list of candidates have equal capacity to serve as a conservation flagship given dedicated marketing efforts” I acknowledge that some simplifications are needed, but this is very unlikely – marketing efforts cannot always overcome people perceptions, due to e.g. cultural background or economic conflicts (even if more perceived than real), as the stories about many large, charismatic, carnivores in developed countries perfectly exemplify.

Response: We point to hard evidence that marketing raises the profiles and willingness of the public to support even the least charismatic species (Verissimo et al, 2017) based on a similar marketing context of international campaigns with a broad target audience. Thus, for example, while some people in the US have very negative perceptions of the wolf, the US public donate millions for lion, tiger and jaguar conservation

Reviewer #4 (Remarks to the Author):

This paper presents a novel analytical approach that could make an important contribution to the field of conservation prioritization. While I did not review previous versions, it seems that the authors were diligent and thorough in addressing prior comments. The authors demonstrate a high degree of spatial overlap in terrestrial regions with high overall biodiversity and those that contain charismatic flagship species commonly used in conservation marketing and fundraising campaigns by NGOs. Their prioritization approach shows that prioritizing areas for conservation based on the presence of flagship species results in the selection of areas with ~80-90% of the total species that would be selected by maximizing only the total number of species. Further, this approach outperforms a random selection of areas in terms of the number of species occurring in selected areas. Their results provide an interesting and useful bridge across the juxtaposition between conservation targeting single, charismatic species (which are often successful fundraisers) and biodiversity conservation.

I found the manuscript at times went beyond what the results of the novel prioritization approach demonstrates. Some of this stems from the writing style and could be addressed with more plain and explicit language. In my opinion terms like ‘conservation benefit’ and ‘biodiversity objective’ can obfuscate what was ultimately analyzed in the paper – the number of species occurring in areas of potential conservation priority that also contain flagship species. More precisely describing these metrics throughout would not only make the manuscript read more easily, but make the methodology more easily understood and convey what the analyses discovered.

Response: Thank you. This is helpful. We have clarified the language throughout. We do retain the term “objective” as it is the appropriate term related to prioritization.

Closely related to this point, conceptually I think it is important to acknowledge and distinguish the difference between overlap in course (100 km²) species presence, and shared characteristics between flagship and background species that would allow the latter to benefit from funding conservation actions directed at the former. The authors should identify and clarify the many intermediate processes that might cause a disconnect between the spatial overlap of species and their ability to mutually benefit from conservation actions (e.g. differences in habitat, shared threats, range overlap, legal protections, etc). The authors are clear in acknowledging that funding raised in the service of a single species and/or it’s habitat cannot be ethically spent on unrelated conservation programs. However, much of the framing and discussion seems to make an implicit assumption that money being directed to an area provides conservation benefit to the species in that area by default.

Response: Please see points 3 and 4 at the beginning of this response.

More specific comments are provided below.

Line 23: This framing, to me, is not quite accurate. The criticism of flagship species is not based on where money goes, but that it is only directed towards single species. I think plenty of conservationists would acknowledge that funding for flagship species is spent in biodiversity rich areas. The criticism is based on what actions those resources fund and whether those ameliorate threats for one or many species. This may seem like a minor

nuance, but the distinction is important to clarify as it underpins the broader framing of this paper.

Response: We have revised the framing in the abstract, which now reads: ... “critics of this species-focused approach argue it wastes resources and often doesn’t benefit broader biodiversity”.

We have also made the conservation context clearer in Line 214 by explaining “The flagship and background species do not need to have overlapping ranges within the priority places, as previous successful fundraising campaigns have simply used flagships to provide examples of the important biodiversity found within a priority region (12).”

Line 27: “Biodiversity objective” is a bit obscure here before any context is given. It might be clearer, if not more accurate, to say something like that you identified places that contain flagship species and maximize biodiversity representation.

Response: Thank you. We have revised the abstract accordingly which now reads:

“Through a novel prioritization approach, we identify places containing flagship species while also maximizing global biodiversity representation (based on 19,616 terrestrial and freshwater species).”

Line 106: I appreciate the desire to frame and discuss this work in a rigorous objective-maximization framework, but I think clarity for the reader could be gained by stating more explicitly the measure you evaluated (i.e. number of background species encompassed) rather than “80 – 89% of the objective.”

Response: This is very helpful. We have revised the language we use to describe our prioritization approach throughout the manuscript.

Line 66: “Here we ask to what extent is fundraising through a flagship species approach a major constraint on delivering efficient place-based conservation at a global scale?” Consider rephrasing this question to more closely align with the outcomes of the optimization analyses. The phrasing at Line 204 would be clearer.

Response- We have revised the initial question to better reflect the language used at 204 in the previous draft. The leading question in the introduction now reads: Here we ask to what degree does requiring the presence of a flagship species influence our ability to achieve place-based conservation objectives at a global scale?

Line 115 – 141: These paragraphs read primarily like methods to me, laying out the steps used to generate different subsets of species and areas to be analyzed. Consider relocating the section to the Methods. At the very least, the subsection title should be changed from ‘the influence of preferential attributes on candidate flagship species and places’ to something more straightforward like ‘flagship species and focal area selection.’

Response: We have moved the steps laying out how the subsets of species and places were generated to the Methods section. We condensed the relevant results into a single paragraph at the beginning of the Results section. We have retitled the section “Flagship species and place-based constraints”

Line 145: “performance efficiency?” Unclear what this refers to, as the optimization approach hasn’t been fully described. It would also seem like one would either measure the performance or the efficiency of the approach, but not ‘performance efficiency?’ I may be misunderstanding if this term is a precise reference to a more specific metric, and if so it should be defined before it appears here.

Response: We have removed “performance” from the sentence and follow it with the explanation of how it is measured.

Lines 135: “To measure efficiency in achieving our objective, we compared the number of background species captured from the places prioritized in our integrated approach with that of the equivalent number of places identified from the place-only and random approaches.”

Line 176: Again, its not clear how you’re measuring efficiency, or what this refers to. Is this area under the species area curves?

Response: In the paragraph starting lines 135-144, we explain what efficiency is and how it is measured. We have restructured this paragraph so our example detailing how efficiency is measured comes before reporting the results. We have added more explanatory text and defer to Table 1 to offer more insight into how efficiency is quantified and compared across scenarios.

Line 148 – 150: While you are clear in defining ‘conservation benefit’ as the number of background species present in a given area, I would prefer a more specific term because of the different factors that would contribute to what most readers would call conservation benefit (i.e. species protections) that can’t be accounted for in the framework addressed in this paper (e.g. habitat/niche overlap between flagship species and background species, commonality of threats, legal authority, etc.)

Response: Agreed. We have removed benefit from this sentence and focus on the number of background species represented as our objective.

Line 204: This strikes me as the clearest, most accurate description of the research question addressed – “whether important places for biodiversity contain flagship species.” I recommend incorporating this phrase, or similar descriptions, when describing the study in preceding sections.

Response: Done. We have rephrased the initial question in the preceding sections along these lines.

Line 235 – 247: This is an important point to acknowledge, that it is conservation actions which ultimately deliver protections for species and conservation benefits. Whether in this paragraph or an additional Discussion paragraph, I think it is important to take this acknowledgement a step further and discuss how the degree of overlap between the available actions that benefit a flagship species and those that benefit background species. Although the degree of overlap between flagship species and background species was measured as presence-absence within 100km² cells, funds raised for flagships will most often be limited to those species range or native habitat. These may have even less overlap with the ranges

and habitats of background species. Some habitats are mutually exclusive, in which case there may be no overlap. If flagship and background species are not threatened by a common set of factors, even spatially overlapping species may not share in the conservation benefits of actions tailored to address threats for flagship species.

Response: Please see point 3 and 4 above.